# Research on Mechanical Properties of Origami Aluminum Honeycomb for Automobile Energy Absorbing Box

**DOI:** 10.3390/ma16010141

**Published:** 2022-12-23

**Authors:** Wei Wei, Fengqiang Zhang, Youdong Xing, Hongxiang Wang, Rongqiang Liu

**Affiliations:** 1School of Mechatronics Engineering, Harbin Institute of Technology, Harbin 150001, China; 2Shenzhen Cansinga Technology Co., Ltd., Shenzhen 518000, China; 3School of Mechanical Engineering, Shandong University of Technology, Zibo 255000, China

**Keywords:** origami aluminum honeycomb, automobile energy absorbing box, mechanical properties, energy absorption, numerical simulation

## Abstract

With the increasing number of automobiles on the road, passive safety has become a particularly important issue. In this paper, an energy-absorbing material, origami aluminum honeycomb, was manufactured by a welding process for use as an automobile energy absorbing box. The mechanical properties and deformation of welded origami aluminum honeycomb in three directions were studied through quasi-static and dynamic compression tests. The results show that the origami aluminum honeycomb had good mechanical energy absorption performance, and the optimal directions are identified. Combined with theoretical analysis, the errors between experiments and simulations are shown. The origami honeycomb structure was designed for use as an automobile energy absorbing box. Analysis shows that it could absorb at least 10% of the kinetic energy of a vehicle during a collision, and could play a role in protecting the interior of the vehicle.

## 1. Introduction

As the number of cars on the road increases, passive safety protection becomes increasingly important. Due to the complexity and high cost of active safety performance, the development of passive safety features to protect passenger safety is a new strategy [1,2,3]. Passive safety is mainly functional when the car is in a collision, the front beam, energy-absorbing box or body can absorb part or all of the energy, thereby reducing the energy transmitted to the passengers [4].

Compared with damage of the car body, it is a better choice to sacrifice energy-absorbing box. Therefore, it is very meaningful to improve the energy-absorbing performance of the energy-absorbing box to protect the safety of vehicles and passengers [5]. Commonly used energy-absorbing materials are generally thin-walled structural materials, such as square tubes and round tubes. They are often used as energy-absorbing structures due to their simple structure and ease of preparation [6,7,8,9].

In addition, emerging porous lightweight materials are increasingly becoming central in energy absorption. Gen [10] designed and fabricated two types of honeycomb-wall curved honeycombs using a fused deposition modeling method by replacing the honeycomb walls of traditional diamond-shaped honeycombs with curved surfaces. Intersecting curved honeycombs with continuous gradients exhibited the best energy-absorption performance compared to conventional honeycombs. SanHa [11] proposed a new bio-inspired hierarchical circular honeycomb (BHCH) that mimics the layered structure of wood. The mechanical properties and energy absorption properties of the proposed structure were investigated by quasi-static compression tests and finite element analysis. Experimental results and parametric studies showed that BHCH was superior to conventional circular honeycomb in terms of energy absorption capacity. The honeycomb structure can be used in a wide range of applications from structural components of defense structures to vehicle protection systems. Yan [12] showed that Nomex honeycombs are widely used for lightweight structures due to their unique porous structure. However, many porous materials are limited by their preparation process, and the price is high. The performance research is still in the laboratory stage and cannot meet the application requirements. Therefore, twe prepared an origami honeycomb structure and describe our analysis of its performance in this paper.

Origami structures have received extensive attention due to their controllable deformation process and excellent energy absorption. The presence of rich crease patterns means that many three-dimensional structures can be fabricated using a variety of sheets, including thin-walled tubes and circular arcs. Some origami structures can also be used as core structures, sandwich panels or arcs, while other such structures can be stacked to form metamaterials [13]. These structures can be designed and tuned through the selection and optimization of appropriate geometric parameters to improve the mechanical properties of the material. Zhai [14] proposed an energy-absorbing structure called a pre-folded honeycomb, which is made from conventional honeycombs with pre-folded traces. The folding mode and energy absorption performance of the pre-folded structure were investigated numerically and experimentally under in-plane impact loading. The results show that the structural strength of the in-plane prefolded honeycomb was nearly eight times that of the conventional honeycomb. Xie [15] studied the mechanical properties of composite structures of stacked multilayer Nomex honeycombs, using a finite element model for building stacked honeycomb composite structures to simulate quasi-static compression. The results were compared with test data to verify the accuracy of the simulation, and both showed that the material had good mechanical energy absorption properties. Gao [16] proposed a novel design of an origami-inspired honeycomb metamaterial with self-locking properties. The self-locking increased the overall energy absorption while keeping the initial low peak load. This work demonstrates a promising approach to exploit the concept of origami to create novel metamaterials with controllable mechanical properties.

The application of origami honeycomb is increasingly extensive, but application of this material in automobile energy absorption has not yet appeared. In this paper, an origami honeycomb material is prepared by welding through the study of automobile energy-absorbing materials. The mechanical properties of the material are analyzed in three directions through quasi-static and dynamic methods. The differences in directional energy absorption are displayed. We further verify the mechanical properties of the welded origami aluminum honeycombs under dynamic conditions, and finally show the feasibility of origami honeycomb materials used as automotive energy-absorbing materials.

## 2. Experiment and Simulation Setting

### 2.1. Experiment Process

Origami aluminum honeycomb (material is 4343 aluminum alloy), was prepared by welding. The preparation process of the test piece used in the test is shown in Figure 1. The origami honeycomb structure is derived from the compression of a hexagonal cylinder. When the tube is compressed, its deformation is affected by the thickness and height of the material and the strain rate, which will form different deformation processes, which is not conducive to the control of material deformation. The origami honeycomb was appropriately improved on this basis. By pre-folding an angle in advance, the honeycomb is deformed according to the predetermined angle. Different from the traditional one-piece forming method, the material prepared in this paper adopts the split forming method, that is, a trapezoidal corrugated long plate with a certain bending angle is punched out first, and then combined and formed by a welding machine.The characteristics of the origami aluminum honeycomb aluminum test samples are shown in Table 1.

The quasi-static test was carried out as follows. Place the origami honeycomb on the universal testing machine in a flat and centered state; set the universal testing machine to compress the specimen at a constant speed of 10 mm/min until it is compacted. Record the morphological changes of the sample during the compression process, and save the test data.

The dynamic test was completed on a drop-weight testing machine with a drop-weight speed of 3 m/s, and the acquisition of relevant data was automatically recorded and saved by the experimental instrument.

### 2.2. Simulation

The simulation process was completed in Ansys LS-DYNA 17.0, the 3D model was generated in Solidworks 2014, the model size was the same as the test size, thin plates were added on the upper and lower sides of the model as pressure plates and bottom plates, and then the model was imported into Ansys LS-DYNA using .IGS format. The parameter settings of origami aluminum honeycomb used the relevant parameters of 4343 aluminum. The upper and lower pressure plates were set as stainless-steel plates, and the parameters of the stainless-steel plates were selected by default [17]. In order to speed up the calculation and reduce the use of computer resources and ensure the convergence of the calculation results, the origami aluminum honeycomb and the upper and lower platens were set to different mesh sizes (Figure 2). Reference [18] shows that when the ratio of mesh size to element size was 0.125 (mesh size/element size = 0.125), the simulation results were in good agreement with the experimental results, so the mesh size of the origami aluminum honeycomb could be deduced from this formula. The pressure plate was set as a solid unit, and the material was set as body self-contact, friction-type, with static friction set to 0.3 and dynamic friction set to 0.2. In [19] it was shown that the quasi-static condition could be satisfied when the loading speed was less than 2 m/s, and when the loading speed was 50 mm/s, the kinetic energy value was much lower than the internal energy. In this part, the lower plate was set to be completely fixed, and the loading speed of the upper platen was 50 mm/s.

## 3. Experimental Process and Results Analysis

### 3.1. Experiment and Analysis of Welded Origami Aluminum Honeycomb Material

#### 3.1.1. Static Results and Analysis

Figure 3 shows pictures of the setup of the welded origami honeycomb test, and Figure 4, Figure 5 and Figure 6 show the three-direction loading process of the material.

When the origami aluminum honeycomb was tested in the lateral direction (Figure 4), the material was in a state of slow compression and deformation. The material first deformed at the top, and then deformed layer by layer. The deformation process was not uniform, showing an irregular shape. There were wavy wrinkles in the corrugated layer. With the increase of the displacement, the material was further deformed until it was completely compacted, and the material did not appear to be laterally widened in the final compression stage.

In the radial compression of the welded origami aluminum honeycomb material, there was no cracking. When the material was compressed, deformation occurred first at the weakest hole, and the hexagonal holes gradually became denser in a similar way to collapse. Along the deformed holes, the material appeared to buckle and deformed gradually along a diagonal line, as shown in the green shapes in Figure 5; further, on the basis of the previous deformation, the upper-left half of the material deformed (green area). Then, the material was in a buckling state as a whole, and the material was dumped to one side until the holes were compacted. Deformation did not occur in the middle part of the material. This is because the middle part of the material was double-layered and was difficult to deform. The deformation occurred at the bend of the trapezoid. Because the plate was bent, deformation was more likely to occur at the bend points, staggering the middle to a more deformable position. The deformation process of the material was not the vertical collapse of the holes; rather, the material was compressed and compacted in the trapezoidal direction after buckling at the weakest point.

The deformation process of the welded origami aluminum honeycomb in the axial direction was as follows (Figure 6). The deformation type of the material was plastic hinge deformation along the half-height cells, as well as folding along the preset inclination angle. The number of plastic hinges was twice the number of cells (a unit is a cell marked in yellow in Figure 6). Deformation first occurred at the weakest inclination in the middle and then the specimen gradually compacted. Observing the front and top views of the compacted material, it can be seen that the deformation process of the material was very uniform, in all cases in the form of plastic hinge folding. The stress–strain energy-absorption diagrams of the material are shown in Figure 7, Figure 8 and Figure 9.

Origami Aluminum Honeycomb 1 (Lateral Direction):

(1) On the whole, the lateral deformation of origami honeycomb 1 was relatively regular, and the middle slope began to deform first. (2) The strength was low, and the curve was relatively stable, showing an upward trend. The reason for this difference from the stable deformation curve is that the material prepared by welding had no cracking phenomena, so when the material had a single-layer compaction, the stress of the material was slow to increase. This is the result of the co-deformation of the single-layer corrugated material and the next layer. (3) The average stress was 0.683 MPa, and the energy absorption was 4005 J.

Origami Aluminum Honeycomb 2 (Radial Direction):

(1) On the whole, the radial deformation of the origami honeycomb was relatively regular. The middle slope first began to deform, and then the deformation began to occur near the upper half of the slope. Finally, the deformation began to occur near the lower half of the slope. After compression, the inclination angle was 50°. (2) The strength was low, the curve fluctuated slightly, and the overall trend was upward. (3) The average strength was 0.677 MPa, and the energy absorption was 4616 J.

Origami Aluminum Honeycomb 3 (Axial Direction):

(1) On the whole, the axial deformation of origami honeycomb 3 was relatively regular. The layer below the middle began to deform first, then the upper layer deformed in a cumulative manner, and finally the structure deformed overall. (2) The strength was high, and the curve fluctuated greatly. (3) The average stress was 2.55 MPa and the energy absorption was 17,813 J. Table 2 shows the statistical results of the static pressure test at room temperature.

#### 3.1.2. Dynamic Results and Analysis

Three-direction dynamic mechanical loading test was carried out on welded origami aluminum honeycombs using drop-weight test equipment. The process and results are as follows.

Under the axial drop weight impact test, the deformation speed of the material was fast. The deformation process (Figure 10) captured by the high-speed camera is shown in the figure. The deformation of the material was mainly plastic hinge folding deformation, the folding position was still at the designed inclination angle, and the folding height was half of the unit cell. The deformation process was similar to that in the static state.

Observing the stress–strain curve of the material during axial dynamic compression (Figure 11), it can be seen that the material had large stress fluctuations, and the initial peak stress was about 8.4 MPa (significantly higher than that of the welded origami aluminum honeycomb during static compression), then the stress continued to fluctuate. The peak value was lower than the initial peak value, the material did not have a smooth plateau stage, and the stress gradually stabilized in the second half until densification occurred.

The material deformed (Figure 12) under high-speed lateral impact, and the deformation process was similar to that under static load. The final state was that the material was compacted more tightly and the deformation was larger under dynamic conditions.

The stress–strain curve of the material during lateral dynamic loading is shown in Figure 13. The initial peak stress value was about 1 MPa, which is much higher than the value under static loading. The curve fluctuates greatly at the plateau stage. This is related to the number of layers of the material. When a single-layer material is deformed under high-speed impact, the fluctuation effect is more severe. The average stress value in the plateau stage was stable at around 0.72 MPa, and then the stress value gradually increased until the densification stage under the dynamic fluctuation of the material.

The deformation mode of the material under radial dynamic loading was similar to that under static load (Figure 14), and the material also demonstrated an inclination phenomenon in the middle deformation, further deforming along the inclined plane until it was fully compacted.

The deformation process of the material under radial dynamic loading was different from that under static loading. The curve of static loading is relatively flat, while the curve of dynamic loading (Figure 15) has large fluctuations. The initial peak stress of the material was 1.8 MPa, which is much higher than that under static loading. After that, the material fluctuated at the plateau stage and the average stress stabilized at around 1.1 MPa until the material densified. Compared with the static densification strain point, the densification point under dynamic loading was significantly larger than that under static load.

Through the three-direction dynamic and static loading analysis of the origami honeycomb material, it can be concluded that in the same direction, the dynamic mechanical performance was better than that under static load; comparing the mechanical properties of the three directions, the axial performance and energy absorption were better than in the other two directions. The deformation and stress under axial loading were further studied by means of simulation and theoretical analysis.

### 3.2. Theoretical Analysis

#### 3.2.1. Axial Deformation

The material was a completely symmetrical structure, and can be regarded as being symmetrical from a single origami cell (green frame interval). When the material was subjected to an axial compression test, the material deformed along the designed angle. The deformation occurred first at the bottom end, and the first plastic hinge appeared. The height of the folding was half the height of the origami cell. As the compression time increased, the whole material was completely compacted along the preset angle, and the number of plastic hinges was the same as the total number of origami cells. The height of the material in Figure 16 is 5 cells, and the number of plastic hinges after complete compaction was 10—that is, two folded plastic hinges deformed after one cell was compressed.

The energy-absorption process of the material was analyzed theoretically. Based on the simplified super-folded unit theory, it can be known that for radial energy absorption deformation, the energy absorption of the material is mainly the plastic deformation energy absorption of each trapezoidal plate, and each trapezoid cell will produce two plastic hinge folds on the left and right. The energy absorption calculation is shown as Formula (1).

Through the above experimental comparison, it can be found that for loading in three directions, the energy-absorption performance of axial loading was the best. For axial compression, the material exhibited plastic buckling deformation, and the buckling position was the position of the folding angle of the material. Each plastic hinge fold corresponded to a layer of folded corner bending.
(1)$$E=M_{0}\theta l,\;M_{0}=\frac{1}{4}\sigma_{0}t_{0}{}^{2},\sigma_{0}=\sqrt{\frac{\sigma_{y}\sigma_{u}}{1+n}},$$
where *σ_y_* and *σ_u_* are yield strength and ultimate strength, respectively, and *n* is 0.2. M_0_ is the full plastic bending moment and *θ* is the bending rotation angle of each hinge line.

*E* is the energy absorption of corrugated folding 10 times. It is concluded that the total energy absorption of the material under the theoretical calculation is 22.2 kJ, compared with the experimental result of 20.2 kJ; the error between the two results is caused by ideal calculation.

#### 3.2.2. Radial Deformation

For the radial deformation process, taking the material unit cell as an example, the deformation process of the material in the ideal state is shown in Figure 17. At this time, based on the simplified super-folded element theory, the calculated energy absorption of the material is 4.8 kJ. The error between experiment and theory was about 3.8%. The deformation process was different from the ideal state because there were two surfaces welded together in the unit cell material. The strength of the two planes was higher than that of the one plane. The deformation was not regular and homogeneous. The structure first deformed on a single layer, resulting in irregular deformation, which is also the reason for the gap between theoretical calculations and experiments.

#### 3.2.3. Lateral Deformation

For the lateral deformation process, taking the material unit cell as an example, the deformation process of the material in the ideal state is shown in the Figure 18. At this time, based on the simplified super-folded element theory, the calculated energy absorption of the material is 3.9 kJ. The error between experiment and theory was about 2.6%. The deformation process of the material was different from the ideal state. In actual working conditions, the material did not display bending and deformation of the hypotenuse on both sides, rather the bending deformation occurred only at one end, and the other end was rotated and deformed. Because the welding was a single-wire welding form, a warped shape appeared at both ends of the wire, resulting in a wavy shape in the final deformation.

### 3.3. Simulation Results and Discussion under Axial Direction

Through the above tests, it can be concluded that the mechanical properties of the material in the axial direction were optimal. Therefore, the axial compression deformation and mechanical properties were studied using simulation.

The axial compression process of the material is shown in (Figure 19). The deformation process of the material under the simulation was similar to the experimental deformation process. The deformation of the material under the simulation first appeared at the corner, then the material continued to fold along the corner until it was completely folded. Observing the stress–strain curve (Figure 20) in the simulation, it can be found that the curve results of the test and the simulation were relatively similar, including the elastic stage, the plateau stage and the compaction stage.

### 3.4. Feasibility Study of Origami Aluminum Honeycomb as an Automotive Energy-Absorbing Element

When a car collides at medium or low speed, in order to ensure the safety of passengers in the cab, the car must absorb part of the impact energy. The kinetic energy of the car is positively related to the weight and speed of the vehicle. The rate of negative acceleration of a car during a crash is very high. The kinetic energy is completely transformed into the deformation energy of the car [20].

It can be seen from the static and dynamic tests that the evaluated material can absorb a large amount of energy within a certain deformation range, and the energy absorbed by the material under dynamic load was larger. However, in order to ensure the safety of passengers, the peak stress should be lower, which is an optimization goal to be pursued in a future work. This study only confirmed the feasibility of the origami aluminum honeycomb material as a vehicle energy-absorbing element. When a car collides, it needs to absorb more than 2000 kJ of energy. In the static load test of the origami aluminum honeycomb, the energy absorption was 22.2 kJ; it absorbed 11.1% of car collision energy, and in the dynamic test the origami aluminum honeycomb absorbed 606.95 kJ, 30.34% of car collision energy (Figure 21). It can be seen from the above data that the origami aluminum honeycomb material could absorb at least 10% of the kinetic energy in a crash, playing a protective role [21].

## 4. Conclusions

In this paper, welded origami aluminum honeycombs were prepared by welding, and the mechanical properties of the honeycombs were analyzed in three directions by static and dynamic methods.

(1)Origami aluminum honeycomb had different deformation processes in three loading directions evaluated. Under the static test, the origami aluminum honeycomb had similar lateral and radial mechanical properties, and the deformation process was also similar. Under quasi-static action, the deformation of the material was flat, and the energy absorption was different after being loaded in three directions. The results show that more energy was absorbed in the axial direction than in the other two.(2)The stress of the material under dynamic loading was higher than that under static loading, and the fluctuation was stronger than that under static loading, indicating that speed has a great impact on the material properties, and similarly the energy absorption under dynamic loading was higher than that under static loading.(3)The mechanical properties of origami aluminum honeycomb were studied experimentally, by simulation and through theoretical analysis. Although there were some errors among the three results, these methods could characterize the accuracy of the material’s energy absorption.(4)The material could absorb a large amount of energy under low- and medium-speed impact. When applied to vehicle energy-absorbing components, it could effectively absorb a minimum of 10% of vehicle impact kinetic energy.

## Figures and Tables

**Figure 1 materials-16-00141-f001:**
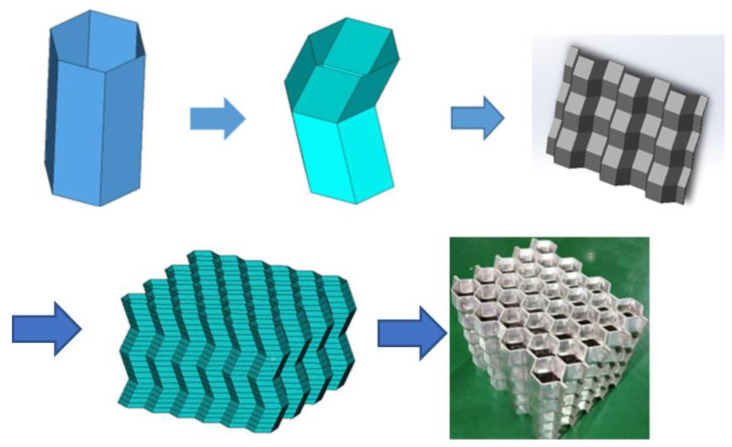
Preparation process of origami aluminum honeycomb material.

**Figure 2 materials-16-00141-f002:**
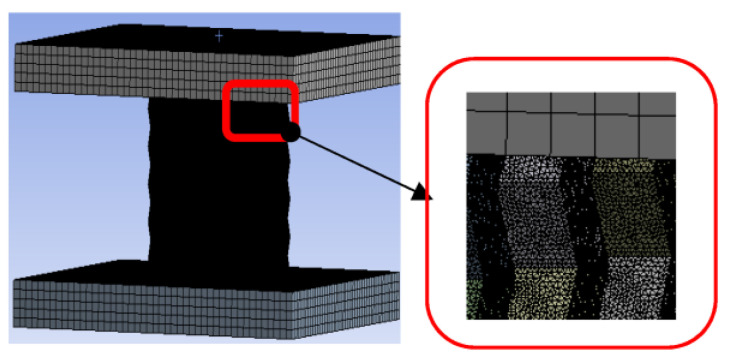
Meshing.

**Figure 3 materials-16-00141-f003:**
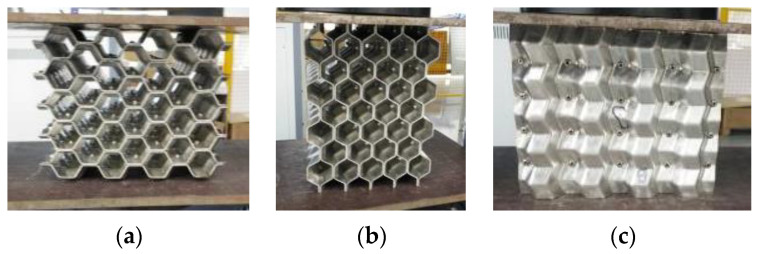
Installation diagram of the welded origami honeycomb test. (**a**) Lateral installation of the test specimen. (**b**) Radial installation of the test specimen. (**c**) Axial installation of the test specimen.

**Figure 4 materials-16-00141-f004:**
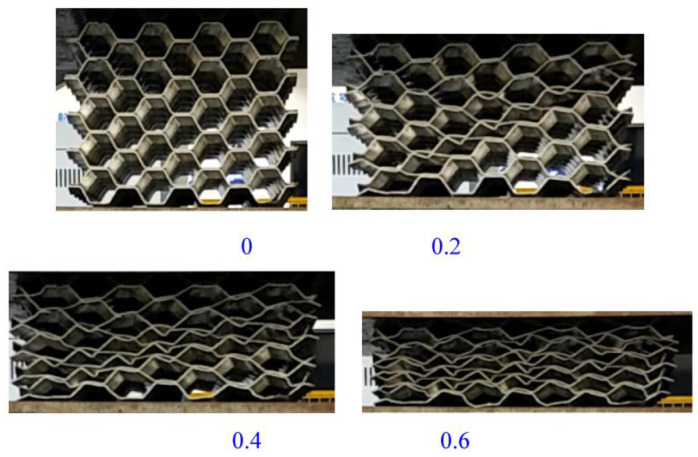
Pictures of the lateral compression process of origami honeycomb 1.

**Figure 5 materials-16-00141-f005:**
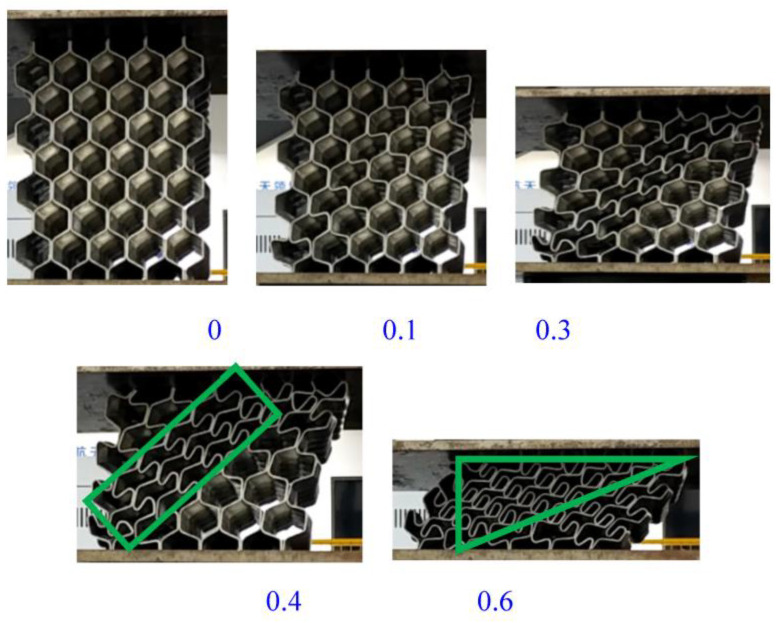
Pictures of the radial compression process of origami honeycomb 2.

**Figure 6 materials-16-00141-f006:**
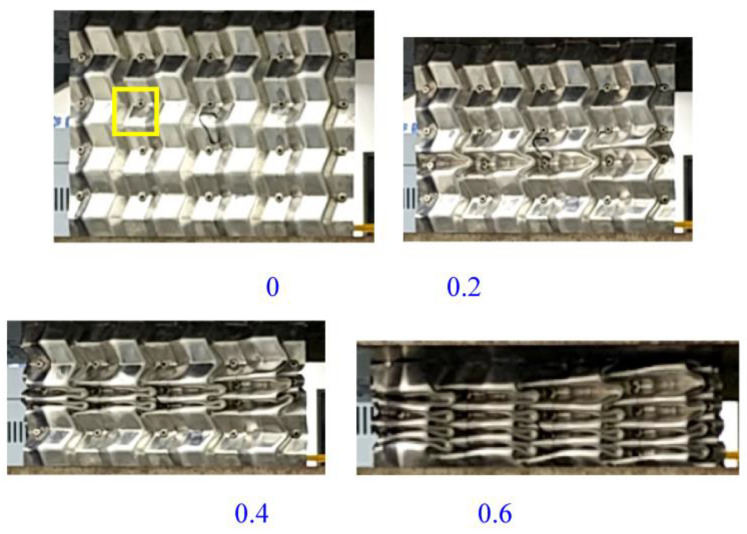
Schematic diagram of the axial compression process of origami honeycomb 3.

**Figure 7 materials-16-00141-f007:**
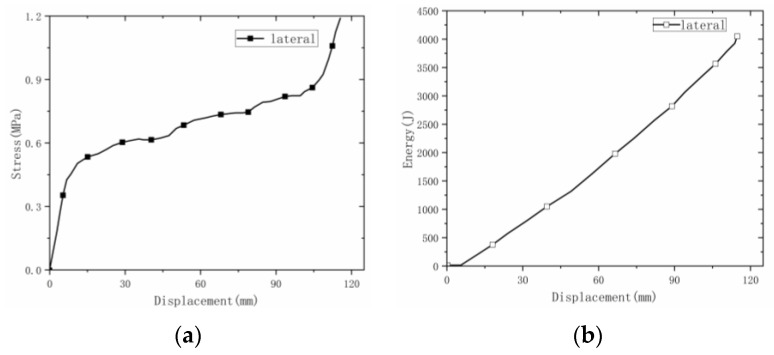
Lateral stress–displacement and energy-absorption curves of welded origami honeycomb 1. (**a**) Stress–displacement curve. (**b**) Energy–displacement curve.

**Figure 8 materials-16-00141-f008:**
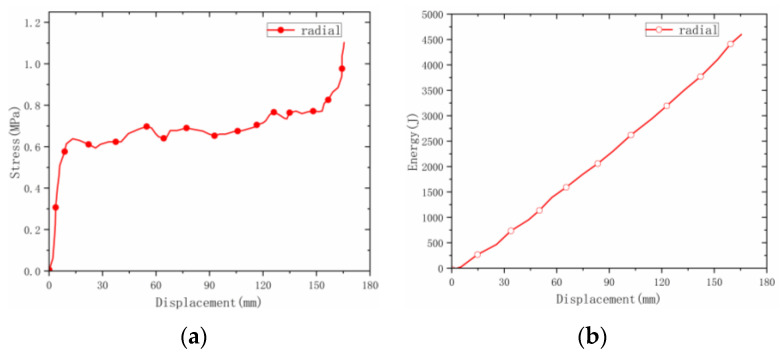
Radial stress–displacement and energy-absorption curves of welded origami honeycomb 2. (**a**) Stress–displacement curve. (**b**) Energy–displacement curve.

**Figure 9 materials-16-00141-f009:**
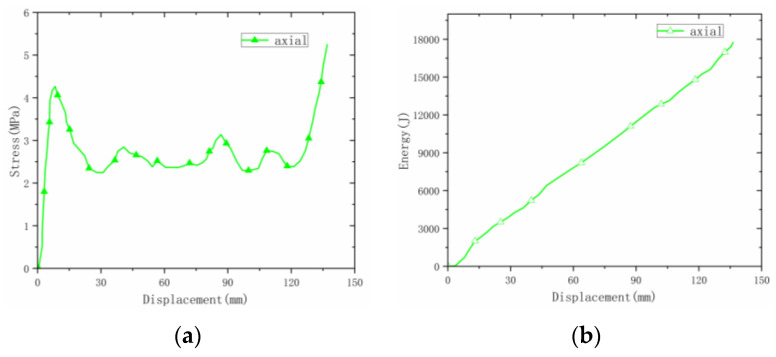
Axial stress–displacement and energy-absorption curves of welded origami honeycomb 3. (**a**) Stress–displacement curve. (**b**) Energy–displacement curve.

**Figure 10 materials-16-00141-f010:**
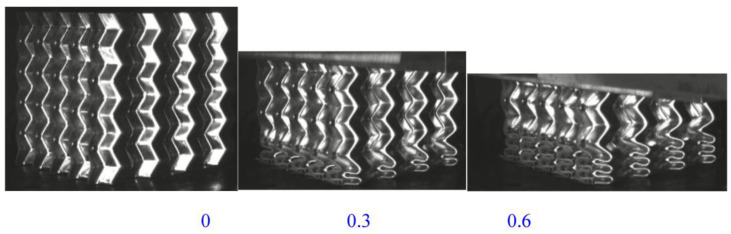
Dynamic test process under axial direction.

**Figure 11 materials-16-00141-f011:**
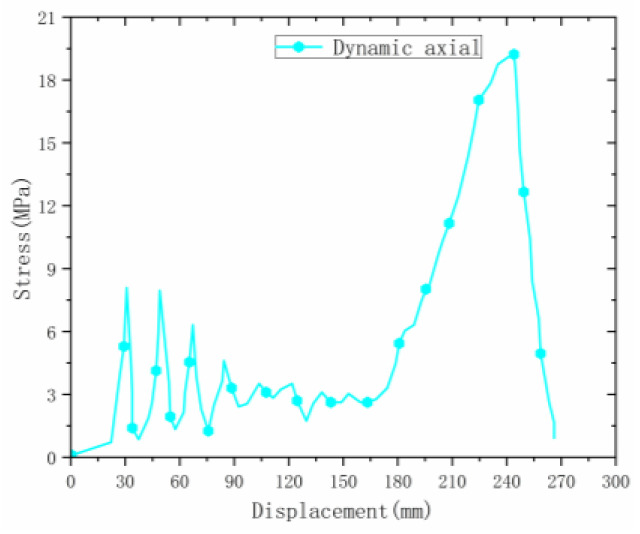
Dynamic test in the axial direction.

**Figure 12 materials-16-00141-f012:**
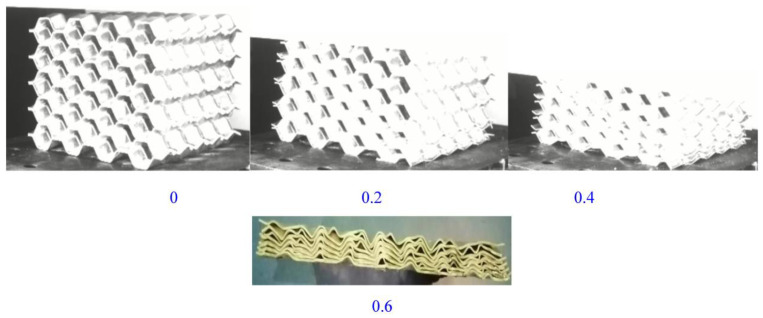
Dynamic test process in the lateral direction.

**Figure 13 materials-16-00141-f013:**
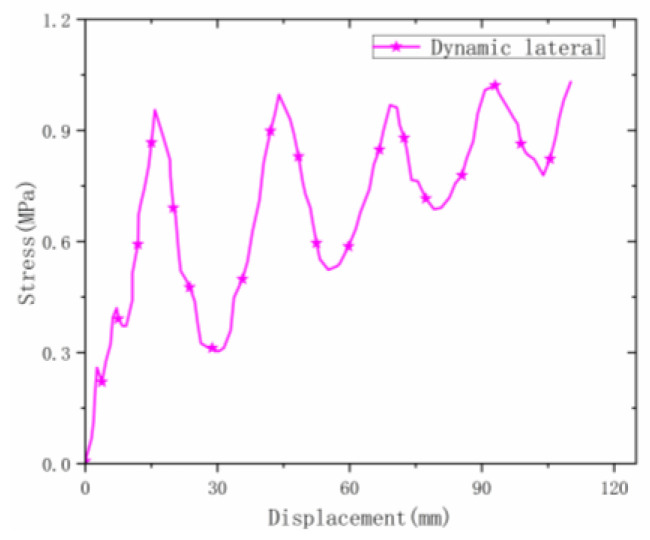
Lateral dynamic stress–strain curve.

**Figure 14 materials-16-00141-f014:**
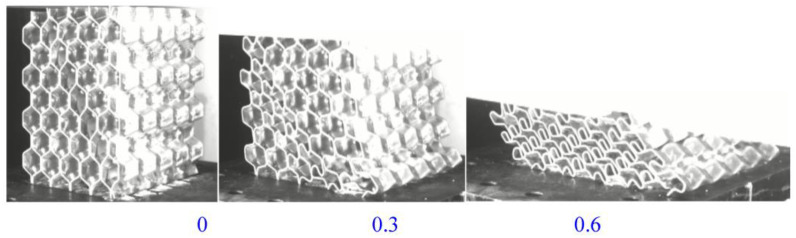
Dynamic test process in the radial direction.

**Figure 15 materials-16-00141-f015:**
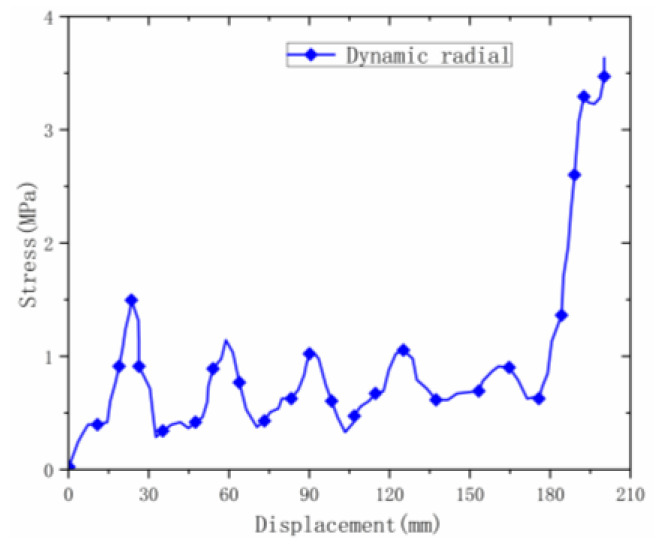
Radial dynamic stress–displacement curve.

**Figure 16 materials-16-00141-f016:**
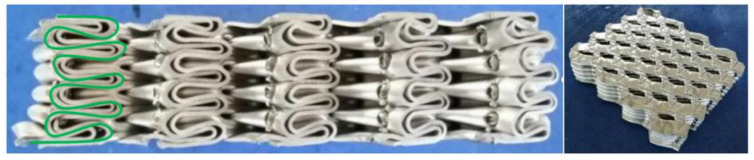
The final configuration of axial deformation.

**Figure 17 materials-16-00141-f017:**
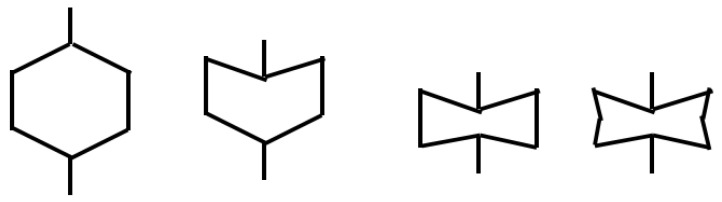
Radial deformation—theoretical analysis.

**Figure 18 materials-16-00141-f018:**
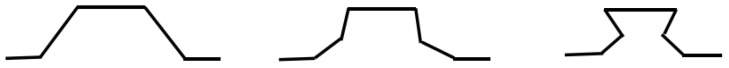
Lateral deformation—theoretical analysis.

**Figure 19 materials-16-00141-f019:**
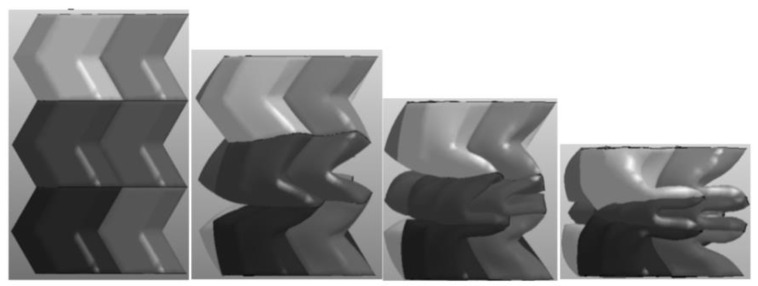
The deformation process under simulation.

**Figure 20 materials-16-00141-f020:**
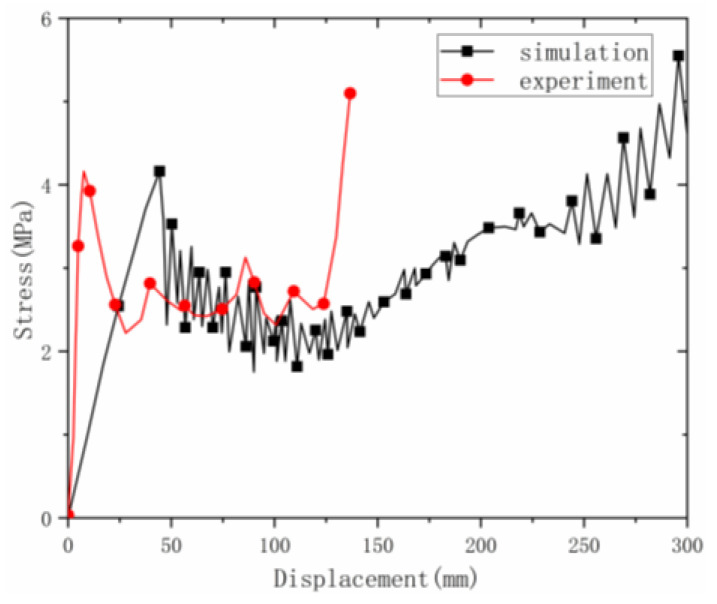
Comparison of simulation and test curves under axial loading.

**Figure 21 materials-16-00141-f021:**
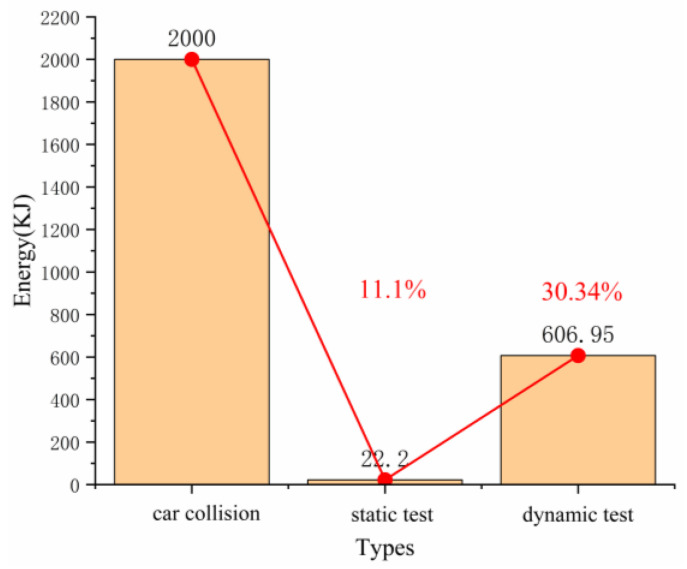
The energy comparison of collision.

**Table 1 materials-16-00141-t001:** The origami aluminum honeycomb test samples.

No.	Specimen	Size (mm)	Initial Height (mm)	ρ (g/cm^3^)
1	Welded origami honeycomb 1 lateral direction	245.4 × 211.1 × 196.8, wall thickness 1.5	196.8	0.2648
2	Welded origami honeycomb 2 radial direction	245.4 × 211.1 × 196.8, wall thickness 1.5	245.4	0.2648
3	Welded origami honeycomb 3 axial direction	245.4 × 211.1 × 196.8, wall thickness 1.5	211.1	0.2648

**Table 2 materials-16-00141-t002:** Welded origami honeycomb data analysis.

No.	Specimen	Energy/Volume (J/cm^3^)	Energy/Mass (J/g)
1	Lateral direction	0.39	1.48
2	Radial direction	0.45	1.71
3	Axial direction	1.74	6.60

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
