# Peer review of "Research on Mechanical Properties of Origami Aluminum Honeycomb for Automobile Energy Absorbing Box"

_materials, 2022, doi:10.3390/ma16010141_

Round 1

Reviewer 1 Report

[1]         Introduction: The objective of the present work carefully.

[2]         Results and Discussion:

Why the author didn’t measure the thermal conductivity, impact and roughness values of the samples?

Why the author didn’t measure the other mechanical properties such as compression strength, Strain, Young’s Modulus, Impact of the samples?

[3]   What does it add to the subject area compared with other published material?

[4]   Fig.7 graph Energy (J) vs. displacement (mm), the author should write the equation of energy absorption?

[5]         References: cite the following recent references

DOI: https://doi.org/10.1088/1742-6596/1795/1/012052

DOI: https://doi.org/10.1088/1742-6596/1795/1/012059

 Best Regards

Author Response

Dear reviewer 1

Tank you very much for your comments, we revised the paper (RESPONSE- Research on mechanical properties  .docx) and respond below(marked red).

Best wishes.

-All authors

Open Review

English language and style

( ) English very difficult to understand/incomprehensible 
( ) Extensive editing of English language and style required 
( ) Moderate English changes required 
(x) English language and style are fine/minor spell check required 
( ) I don't feel qualified to judge about the English language and style 

Comments and Suggestions for Authors

  • Introduction: The objective of the present work carefully.

Response: Thanks to the comments. We revised the introduction carefully, and marked red in the paper. Thanks again.

[2]         Results and Discussion:

Why the author didn’t measure the thermal conductivity, impact and roughness values of the samples?

Response: Thanks to the comments. The origami honeycomb aluminum in this paper mainly studies the mechanical properties of compression in three directions. We obtained the optimal direction, and aim to be used as an energy absorbing device for automobiles. For heat conduction, this material has not been studied, and the heat conductivity plays a small role in collision energy absorption. The base material of origami honeycomb aluminum is commercially available aluminum, which is prepared by welding process. The origami honeycomb is mainly used for energy absorption, and its compression mechanical property is our focus point.

Why the author didn’t measure the other mechanical properties such as compression strength, Strain, Young’s Modulus, Impact of the samples?

Response: Thanks to the comments. As an energy absorbing material, we mainly care about its stress-strain curve. The energy absorption of the material is determined by the area enclosed by the stress-strain curve. To some extent, as an energy absorbing material, the peak stress should be as small as possible. The main point of this paper is the application feasibility of the material, not the performance of the material itself.

[3]   What does it add to the subject area compared with other published material?

Response: Thanks to the comments. In this paper, the mechanical properties of origami honeycomb aluminum material in three directions are studied, and the objective is to apply it to automobile shock absorbers as energy absorption materials.In paper[Geometry design and in-plane compression performance of novel origami honeycomb material], [Energy absorption of pre-folded honeycomb under in-plane dynamic loading]etal, the mechanical response of the structure size to the origami honeycomb is more studied than the specific application.

  • 7 graph Energy (J) vs. displacement (mm), the author should write the equation of energy absorption?

Response: Thanks to the comments. The equation of energy ansorption is  ,We have added in the paper that energy absorption represents the area enclosed between the force and displacement curve. In the paper, we directly give the energy -displacement curve.

[5]         References: cite the following recent references

DOI: https://doi.org/10.1088/1742-6596/1795/1/012052

DOI: https://doi.org/10.1088/1742-6596/1795/1/012059

Response: Thanks to the comments. We cite the paper[An investigation of the Structural, Electrical and Optical Properties of Graphene-Oxide Thin Films Using Different Solvents] [Solid State Reaction Synthesis and Characterization of Cu doped TiO2 Nanomaterials] in the paper.

 Best Regards

Thank you very much for your comments.

Best wishes.

All authors

Reviewer 2 Report

The authors manufactured an energy-absorbing material, origami aluminum honeycomb welding process. The mechanical properties and deformation of welded origami aluminum honeycomb in three directions were studied through quasi-static and dynamic compression tests. The results show that the origami honeycomb aluminum has good mechanical energy absorption performance, and the optimal direction and reason as an energy-absorbing and shock-absorbing material are pointed out. Combined with theoretical analysis, the reasons for the errors between experiments, simulations and theory are showed. The origami honeycomb structure is designed to be used as a collision energy-absorbing material on the vehicle. Analysis shows that it can absorb at least 10% of the kinetic energy of the vehicle during a collision, and play a role in protecting the interior of the vehicle.

The paper will be ready for publication after major revision.

Please highlight your contributions in introduction.

Some figures should be replotted and edited. Use times new romans fonts, clear colors, and 400 DPI resolution. 

” During lateral dynamic loading, the stress value (Figure 13) becomes larger…………….”, Revise this paragraph. 

The manuscript should start by a strong paragraph. 

The format of references is not suitable for the journal.

Use Mendeley or Endnote to fix the references format.

The abstract should be rewritten to reflect the significance of the proposed work. The current abstract shows a lot of background information. 

Conclusion: What are the advantages and disadvantages of this study compared to the existing studies in this area?

Experimental design should be discussed.

Support the discussion  and the introduction by the following”

Bistable Morphing Composites for Energy-Harvesting Applications

This may help the authors to understand  and the importance of this subject

The inspiration of your work must further be highlighted. Some suggested recent literatures should add.  Add future works as bullets.

Looking and wishes for the revised version.

Author Response

Dear reviewer 2

Tank you very much for your comments, we revised the paper (RESPONSE- Research on mechanical properties  .docx) and respond below(marked blue).

Best wishes.

-All authors

Open Review

(x) I would not like to sign my review report

( ) I would like to sign my review report

English language and style

( ) English very difficult to understand/incomprehensible

( ) Extensive editing of English language and style required

( ) Moderate English changes required

(x) English language and style are fine/minor spell check required

( ) I don't feel qualified to judge about the English language and style

Comments and Suggestions for Authors

The authors manufactured an energy-absorbing material, origami aluminum honeycomb welding process. The mechanical properties and deformation of welded origami aluminum honeycomb in three directions were studied through quasi-static and dynamic compression tests. The results show that the origami honeycomb aluminum has good mechanical energy absorption performance, and the optimal direction and reason as an energy-absorbing and shock-absorbing material are pointed out. Combined with theoretical analysis, the reasons for the errors between experiments, simulations and theory are showed. The origami honeycomb structure is designed to be used as a collision energy-absorbing material on the vehicle. Analysis shows that it can absorb at least 10% of the kinetic energy of the vehicle during a collision, and play a role in protecting the interior of the vehicle.

The paper will be ready for publication after major revision.

Please highlight your contributions in introduction.

Response: Thanks to the comments. In the introduction, we have marked our main work in blue. Here, it is described as follows. We study the mechanical properties of origami honeycomb in three directions, and aim to apply it as energy absorbing parts of automobiles. Through further analysis, we can ensure its feasibility.

Some figures should be replotted and edited. Use times new romans fonts, clear colors, and 400 DPI resolution.

Response: Thanks to the comments. We have made appropriate modifications, thank you again.

” During lateral dynamic loading, the stress value (Figure 13) becomes larger…………….”, Revise this paragraph.

Response: Thanks to the comments. We have revised, and the modified contents are as follows. [Lateral dynamic loading is shown as Figure 13. The initial peak stress value is about 1 MPa, which is much higher than the value of static loading. The material fluctuates greatly at the plateau stage. The reason is related to the number of layers of the material. When the single-layer material is deformed under the high-speed impact state, the fluctuation effect is more severe. The average stress value in the plateau stage is stable at around 0.72 MPa, and then the stress value gradually increases until the densification stage under the dynamic fluctuation of the material].

The manuscript should start by a strong paragraph.

Response: Thanks to the comments. We have revised the abstract and introduction.

The format of references is not suitable for the journal.

Response: Thanks to the comments. We have revised the references.

Use Mendeley or Endnote to fix the references format.

Response: Thanks to the comments. We have revised the references.

The abstract should be rewritten to reflect the significance of the proposed work. The current abstract shows a lot of background information.

Response: Thanks to the comments. We have revised the abstract and introduction.

Conclusion: What are the advantages and disadvantages of this study compared to the existing studies in this area?

Response: Thanks to the comments. In this paper, the mechanical properties of origami honeycomb aluminum material in three directions are studied, and the goal is to apply it as automobile shock absorbers.In paper [Geometry design and in-plane compression performance of novel origami honeycomb material] [Energy absorption of pre-folded honeycomb under in-plane dynamic loading], more attention was paid to the mechanical response of structure size to origami honeycomb, and no specific application was mentioned.

Experimental design should be discussed.

Response: Thanks to the comments. We have described the experiment process and added it to the paper.

Place the origami honeycomb on the universal testing machine in a flat and centered state, and the universal testing machine compresses the specimen at a constant speed of 10 mm/min until it is compacted. Record the morphological changes of the sample during the compression process, and save the test data.

The dynamic test was completed on a drop-weight testing machine with a drop-weight speed of 3 m/s, and the acquisition of relevant data was automatically recorded and saved by the experimental instrument.

Support the discussion  and the introduction by the following”

Bistable Morphing Composites for Energy-Harvesting Applications

This may help the authors to understand  and the importance of this subject

Response: Thanks to the comments. We have studied the relevant references and revised the paper in detail.

The inspiration of your work must further be highlighted. Some suggested recent literatures should add.  Add future works as bullets.

Response: Thanks to the comments. We have studied and cited relevant references.

Looking and wishes for the revised version.

Thank you very much for your comments.

Best wishes.

All authors

Reviewer 3 Report

The authors did not mention the limitations and future scope of this work.

Were the experiments performed for the results shown in Figure 21? If yes, how? Elaborate on them in detail.

Details about the simulation performed to verify the results shown in Figure 19 are missing. What were the boundary conditions?

In the same figure, there is no displacement after 120 - 125 mm for experimentation. Justify? here, details about the experimentation are also missing.

What are the plateau stage and compaction stage? Use standard words.

Figure 18 shows lateral deformation using theoretical analysis. Where are the details of this analysis? Figure 18 shows some shapes. Where are the dimensions and labeling? The same is true for Figure 17. Where are the details of this analysis?

How to compare deformation in different directions?

What is the scope of additive manufacturing here?

The quality of the graphs is very poor.

The author's main focus is automobile energy absorption. For which applications within automotive domains is this work suitable?

Comment on Crush Strength Range.

The simulation images can be added regarding the Displacement characteristics of the honeycomb for the L, T, and W direction tests.

Author Response

Dear reviewer 3

Thanks for your comments, the response are shown as follows, in addition, we revised the paper carefully (marked red).

Best wishes.

All authors.

Round 2

Reviewer 2 Report

Accept.

Author Response

Dear reviewer:

Thank you very much for your comments, we revised our paper and sent back.

Best wishes.

Reviewer 3 Report

The authors tried addressing my comments.